# Effects of smoking cessation using varenicline on the serum concentrations of oxidized high-density lipoprotein: Comparison with high-density lipoprotein cholesterol

**Akira Umeda**[1]*, **Kazuya Miyagawa**[2], **Atsumi Mochida**[2], **Hiroshi Takeda**[2], **Yoshiyuki Ohira**[3], **Toru Kato**[4], **Yasumasa Okada**[5], **Kazuhiko Kotani**[6]

1 Department of General Medicine, School of Medicine, International University of Health and Welfare (IUHW), IUHW Shioya Hospital, Yaita, Japan, 2 Department of Pharmacology, School of Pharmacy, International University of Health and Welfare, Otawara, Japan, 3 Department of General Medicine, School of Medicine, International University of Health and Welfare, Narita, Japan, 4 Department of Clinical Research, National Hospital Organization Tochigi Medical Center, Utsunomiya, Japan, 5 Department of Internal Medicine, National Hospital Organization Murayama Medical Center, Musashimurayama, Japan, 6 Division of Clinical Laboratory Medicine, Jichi Medical University, Shimotsuke-City, Japan

* umeda@iuhw.ac.jp

**Data Availability Statement:** All relevant data are within the paper and its Supporting information files.

## Abstract

### Background

The oxidized high-density lipoprotein (oxHDL) is a possible marker for cardiovascular diseases. This study investigated the effects of smoking cessation with varenicline (a partial agonist of nicotinic acetylcholine receptors) on the levels of oxHDL in the serum of subjects compared with those of high-density lipoprotein cholesterol (HDL-C).

### Methods

Data of 99 nicotine-dependent adult subjects who visited the smoking cessation outpatient services at International University of Health and Welfare Shioya Hospital were reviewed. Each subject was treated with varenicline titrated up to 1.0 mg twice daily for 12 weeks. Serum levels of oxHDL and HDL-C were repeatedly measured by enzyme-linked immunosorbent assay and enzymatic method, respectively.

### Results

The serum levels of oxHDL were significantly decreased from 163.2 ± 96.6 to 148.3 ± 80.7 U/mL ($p = 0.034$, n = 99). This effect was more prominent when the data of subjects in whom the treatment was objectively unsuccessful (exhaled carbon monoxide at 3 months $\geq$ 10 ppm) were omitted (from 166.6 ± 98.4 to 147.4 ± 80.6 U/mL; $p = 0.0063$, n = 93). In contrast, the serum levels of HDL-C were significantly increased ($p = 0.0044$, n = 99). There was a close relationship between the baseline levels of oxHDL and HDL-C (R = 0.45, $p < 0.0001$, n = 99). Changes in the levels of oxHDL were closely associated with changes in the levels of exhaled carbon monoxide in subjects in whom smoking cessation with

**Funding:** K.K. This work was partly supported by a MEXT KAKENHI Grant (No. JP 19K07872). This funder had no role in study design, data collection and analysis, decision to publish, or preparation of the manuscript.

varenicline was very effective (decrease in exhaled carbon monoxide by $\geq$ 15 ppm after treatment with varenicline; R = 0.42, $p$ = 0.0052, n = 43).

## Conclusions

Although there was a close relationship between the baseline serum concentrations of oxHDL and HDL-C, smoking cessation decreased oxHDL and increased HDL-C. This effect on oxHDL may be associated with the effectiveness of smoking cessation.

## Introduction

Atherosclerosis, arterial stenosis, and cardiovascular diseases are leading causes of death [1, 2]. Approximately 11–13% of cardiovascular deaths are attributed to smoking [1, 2]. It is widely thought that high-density lipoprotein cholesterol (HDL-C) exerts protective effects on the cardiovascular system [3, 4]. Recently, it was reported that HDL-C and whole high-density lipoprotein (HDL) prevent inflammation and oxidative stress and promote cholesterol efflux from arterial walls, thus reducing the formation of lesions [5–8]. This cholesterol efflux capacity is inversely associated with the prevalence of obstructive coronary artery disease and the incidence of cardiovascular events [5–8]. It has been reported that the concentration of HDL-C is low in smokers [9, 10]. An increase in HDL-C through lifestyle changes (e.g., smoking cessation, physical exercise) is thought to be beneficial for cardiovascular health [10].

The main component of HDL, apolipoprotein A-1 (apoA-I), is easily oxidized. Oxidization of HDL renders it dysfunctional or even pathogenic [11, 12]. Oxidized high-density lipoprotein (oxHDL) is associated with coronary arterial spasm [13]. The serum concentration of oxHDL has been positively associated with coronary artery calcification [14]. It has been reported that oxHDL is increased in patients with diabetes, and is predictive of cardiovascular disease outcomes in patients undergoing hemodialysis for chronic renal failure [15, 16]. OxHDL is also linked to increased plasma glucose levels in patients with non-diabetic dyslipidemia [11]. Non-alcoholic fatty liver disease may be associated with oxHDL [17, 18]. Moreover, He et al. demonstrated that apoA-I mimetic peptide 4F inhibited oxHDL-induced dysfunction of endothelial repairing in a mice model of vascular electric injury [19].

A preliminary report suggested that the levels of oxHDL are higher in smokers versus non-smokers [20]. However, changes in the levels of oxHDL in the blood of subjects who attempt smoking cessation have not been evaluated. In this study, we compared the serum concentration of oxHDL in nicotine-dependent subjects before and after smoking cessation with varenicline, a partial agonist of the α4β2 nicotinic acetylcholine receptor [21, 22]. Furthermore, we evaluated correlations between baseline data of oxHDL and various markers. Similarly, we examined correlations between changes in the serum concentration of oxHDL (ΔoxHDL) and various markers. Data of oxHDL and HDL-C were also compared.

In addition, self-reporting at the smoking cessation outpatient service is occasionally inaccurate. The concentration of exhaled carbon monoxide (CO) is an objective marker of smoking behavior; however, some subjects in whom the treatment was successful (according to self-reports) showed high levels of CO, which could indicate incompleteness [23]. Furthermore, small decreases in exhaled CO could indicate both success in relatively light smokers and incompleteness in any type of smokers with regard to smoking cessation. Considering these findings, we performed some additional subgroup analyses focusing on the effectiveness of this treatment.

## Methods

### Subjects

This study was approved by the Ethics Committee of the International University of Health and Welfare (IUHW, approval number 13-B-62). All subjects provided written informed consent prior to their participation in this study in the form of opt-out on the website [24]. The inclusion criteria were: age $\geq$ 20 years; Brinkman index (smoking index) $\geq$ 200; Tobacco Dependence Screener $\geq$ 5, and stated motivation to quit smoking [25]. Enrollment to this study was conducted from May 2015 to July 2020 at the smoking cessation outpatient service of IUHW Shioya Hospital. Stored samples and information of subjects from other previous investigations [26, 27] were used.

### Study procedures

Each subject was treated with varenicline (CHANTIX®, Pfizer Inc., New York, NY, U.S.A.) titrated up to 1.0 mg twice daily (0.5 mg once daily for 3 days, followed by 0.5 mg twice daily for 4 days, and 1.0 mg twice daily for 11 weeks) (Fig 1). The target smoking cessation date was planned to be 7 days after the initiation of treatment with varenicline (i.e., day 8). Blood was sampled at baseline (before taking the first varenicline) and at 3 months. The concentration of exhaled CO was measured at each visit using a Micro CO Monitor (Micro Medical Ltd., Rochester, United Kingdom) [23]. Varenicline was used for 84 days. Subjects in whom the treatment was successful, according to self-reporting, were defined as those with zero use of tobacco for $\geq$ 2 months. Subjects in whom the treatment was objectively unsuccessful were defined as those with exhaled CO at 3 months $\geq$ 10 ppm. Subjects in whom treatment was very effective were defined as those with a decrease in exhaled CO by $\geq$ 15 ppm after treatment with varenicline. Data of subjects who quit the cessation process were excluded.

### Measurement of oxHDL and HDL-C

The serum levels of oxHDL and HDL-C were repeatedly measured using an enzyme-linked immunosorbent assay and an enzymatic method, respectively [11, 28]. As for oxHDL, the monoclonal antibody against oxidized apoA-I and a biotinylated anti-human apoA-I monoclonal antibody (Ikagaku Co. Ltd., Japan) were reacted in the microplates for each serum sample. Originally, the assay of oxHDL used in the study was based on the $H_2O_2$-induced oxidative modification of apoA-I [11]. The intra- and inter-assay coefficient variations were respectively 8.2 and 10.0% [11].

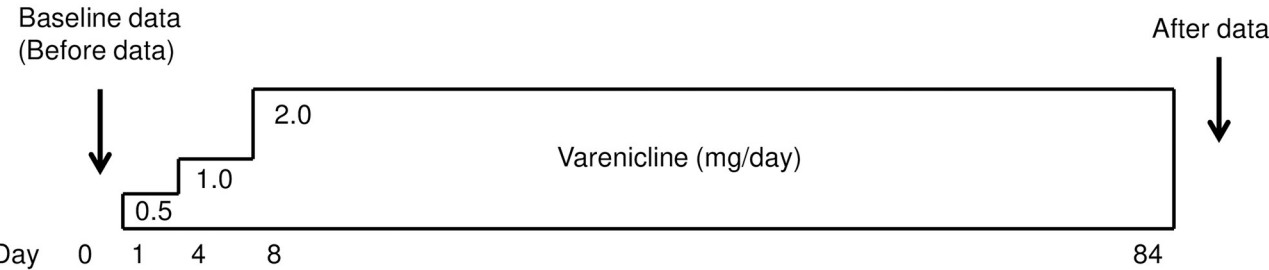

**Fig 1. Protocol (timeline) of this study.** All the subjects (n = 99) were treated with varenicline for 84 days. Each subject was treated with varenicline titrated up to 1.0 mg twice daily (0.5 mg once daily for 3 days, followed by 0.5 mg twice daily for 4 days, and 1.0 mg twice daily for 11 weeks). The target smoking cessation date was planned to be 7 days after the initiation of treatment with varenicline (i.e., day 8). Data were obtained at baseline (before taking the first varenicline) and at 3 months.

### Routine blood tests in clinical practice

Biochemical blood analyses were performed using a TBA-c16000 device (Canon Medical Systems Corporation, Otawara, Japan). Complete blood counts and hemogram parameter measurements were performed using a CELL-DYN Sapphire device (Abbott Diagnostics, Santa Clara, CA, U.S.A.). The levels of hemoglobin A1c and blood glucose were measured using ADAMS A1c HA-8182 (ARKRAY, Inc., Kyoto, Japan) and ADAMS Glucose GA-1170 (ARKRAY, Inc.) devices, respectively.

### Data analysis

Data are shown as the mean ± standard deviation, unless otherwise indicated. The before-after change data of Item A (termed ΔItem A) was calculated as follows: Item A at 3 months after the initiation of smoking cessation with varenicline minus Item A immediately before the initiation of treatment. Student's unpaired *t*-test was used for the comparison between subjects who self-reported successful or incomplete smoking cessation (two-tailed). Student's paired *t*-test was used for the comparison between baseline and after smoking cessation (two-tailed). Spearman's regression analysis was used to evaluate the univariate relationship. Using the forced entry method, multiple regression analysis was performed with the four explanatory variables (ΔoxHDL, ΔHDL-C, age, and sex) to obtain the objective variable of ΔCO in the population of subjects in whom smoking cessation with varenicline was very effective. The Excel Statistics software, 2010 version (Social Survey Research Information Co., Ltd., Tokyo, Japan) was used for the analysis. Statistical significance was set at $p < 0.05$.

## Results

### Study population

The data of 99 nicotine-dependent subjects (79 males, 20 females; mean age: 59.1 ± 11.5 years) who visited the smoking cessation outpatient services at IUHW Shioya Hospital from 2010 to 2020 were reviewed. All the subjects (n = 99) were treated with varenicline for 84 days. Demographics of enrolled smokers are shown in Table 1. The measurement of oxHDL was initiated in 2015. For subjects who visited the hospital before 2014, frozen serum samples and clinical data were used. There were no significant differences between the initial profiles of subjects with successful and incomplete treatment based on the self-reports.

### Effects of smoking cessation with varenicline on serum oxHDL and HDL-C

The serum levels of oxHDL were significantly decreased from 163.2 ± 96.6 to 148.3 ± 80.7 U/mL ($p = 0.034$, n = 99) after smoking cessation with varenicline (Fig 2A). In contrast, the

**Table 1. Demographics of smokers who participated in this study.**

|  | All subjects | Successful | Incomplete | *p*-value |
|---|---|---|---|---|
| n | 99 | 82 | 17 | |
| Males/females (n) | 79/20 | 65/17 | 14/3 | |
| Age (years) | 59.1±11.5 | 59.5±11.4 | 57.6±12.1 | 0.56 |
| Pack-years | 42.3±21.6 | 42.5±20.6 | 43.1±25.6 | 0.92 |
| Current smoking (cigarettes/day) | 21.7±7.9 | 21.9±8.3 | 20.5±5.8 | 0.49 |
| Exhaled carbon monoxide (ppm) | 22.4±12.8 | 22.8±13.0 | 20.3±11.7 | 0.50 |
| Nicotine dependence score (Tobacco Dependence Screener) | 7.5±1.7 | 7.5±1.7 | 7.4±1.9 | 0.79 |

Successful or incomplete status was determined according to self-reporting information ("successful" was defined as 0 cigarette/day for ≥ 2 months). Data (age, sex, pack-years, cigarettes/day, expired carbon monoxide, and nicotine dependence score) were obtained on the first visit (mean ± standard deviation).

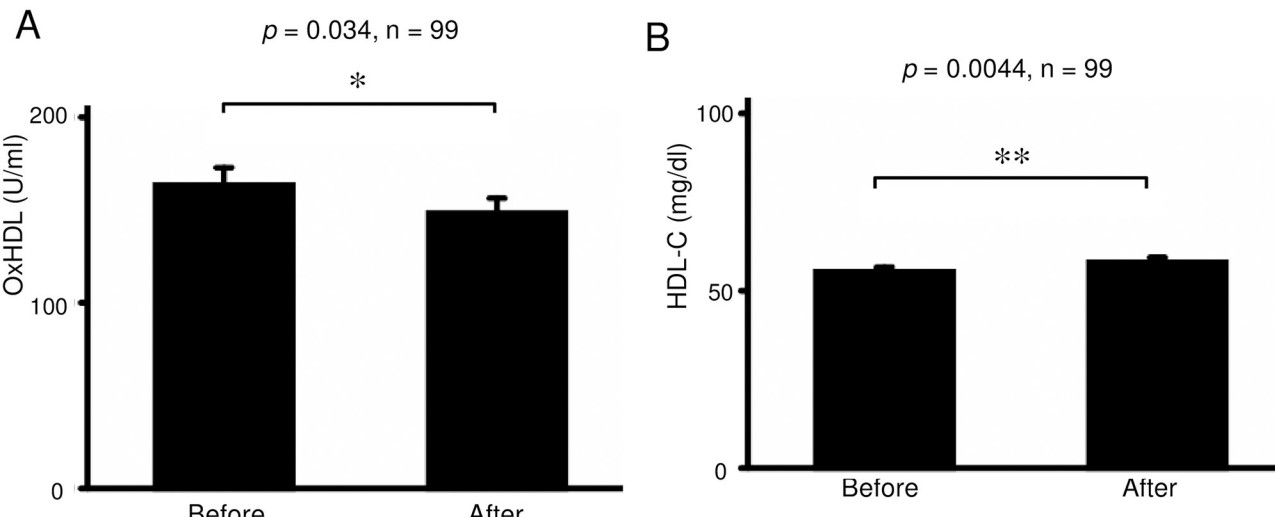

**Fig 2. Effects of smoking cessation with varenicline on the serum levels of oxHDL and HDL-C in nicotine-dependent subjects.** (A) OxHDL was significantly decreased after smoking cessation with varenicline ($p$ = 0.034, n = 99). (B) HDL-C was significantly increased after smoking cessation with varenicline ($p$ = 0.0044, n = 99). $^*$ $p < 0.05$, $^{**}$ $p < 0.01$. Bars: SEM. Abbreviations: HDL-C, high-density lipoprotein cholesterol; oxHDL, oxidized high-density lipoprotein.

serum levels of HDL-C were significantly increased after smoking cessation with varenicline (from 55.3 ± 14.3 to 57.9 ± 14.9 mg/dl; $p$ = 0.0044, n = 99) (Fig 2B). These effects were similarly seen in successful subgroup, according to self-reporting (p = 0.032 and 0.0068, respectively, n = 82) (Fig 3A and 3B), but were insignificant in incomplete subgroup ($p$ = 0.63 and 0.39, respectively, n = 17) (Fig 3C and 3D). This effect on oxHDL was more prominent when the data of subjects in whom the treatment was objectively unsuccessful (exhaled carbon monoxide [CO] at 3 months ≥ 10 ppm) were omitted (from 166.6 ± 98.4 to 147.4 ± 80.6 U/mL; $p$ = 0.0063, n = 93). (Fig 4A). In contrast, this $p$-value on HDLC did not decrease by this omission (from 56.0 ± 14.5 to 58.4 ± 15.2 mg/dl; $p$ = 0.011, n = 93) (Fig 4B).

## Relationship between oxHDL and HDL-C

The baseline data of serum oxHDL and HDL-C showed a close correlation (R = 0.47, $p < 0.0001$, n = 99) (Fig 5A). Changes in these parameters are shown as arrow blots in Fig 5B. The patterns of change varied, but the vector of average changes was a decrease in oxHDL and an increase in HDL-C (from the red circle to the yellow circle in Fig 5B).

## Changes in other parameters and relationships with oxHDL or HDL-C

Body weight and body mass index were clearly increased after smoking cessation with varenicline ($p < 0.001$, respectively) (Fig 6). The use of tobacco and exhaled CO were also markedly decreased ($p < 0.001$ for both). Significant increases in parameters associated with weight increase, such as γ-glutamyl transpeptidase and hemoglobin A1c, were found ($p < 0.01$ and $p < 0.001$, respectively). The levels of sodium ion were significantly decreased ($p < 0.01$). Baseline blood urea nitrogen (BUN) showed a positive correlation with both baseline data of oxHDL and HDL-C (R = 0.37 and 0.31, respectively). Baseline triglyceride (TG) showed a negative correlation with baseline oxHDL and HDL-C (R = −0.26 and −0.45, respectively). There were no significant relationships between the baseline serum levels of oxHDL and baseline data of exhaled CO, pack-years, or serum C-reactive protein levels in nicotine-dependent

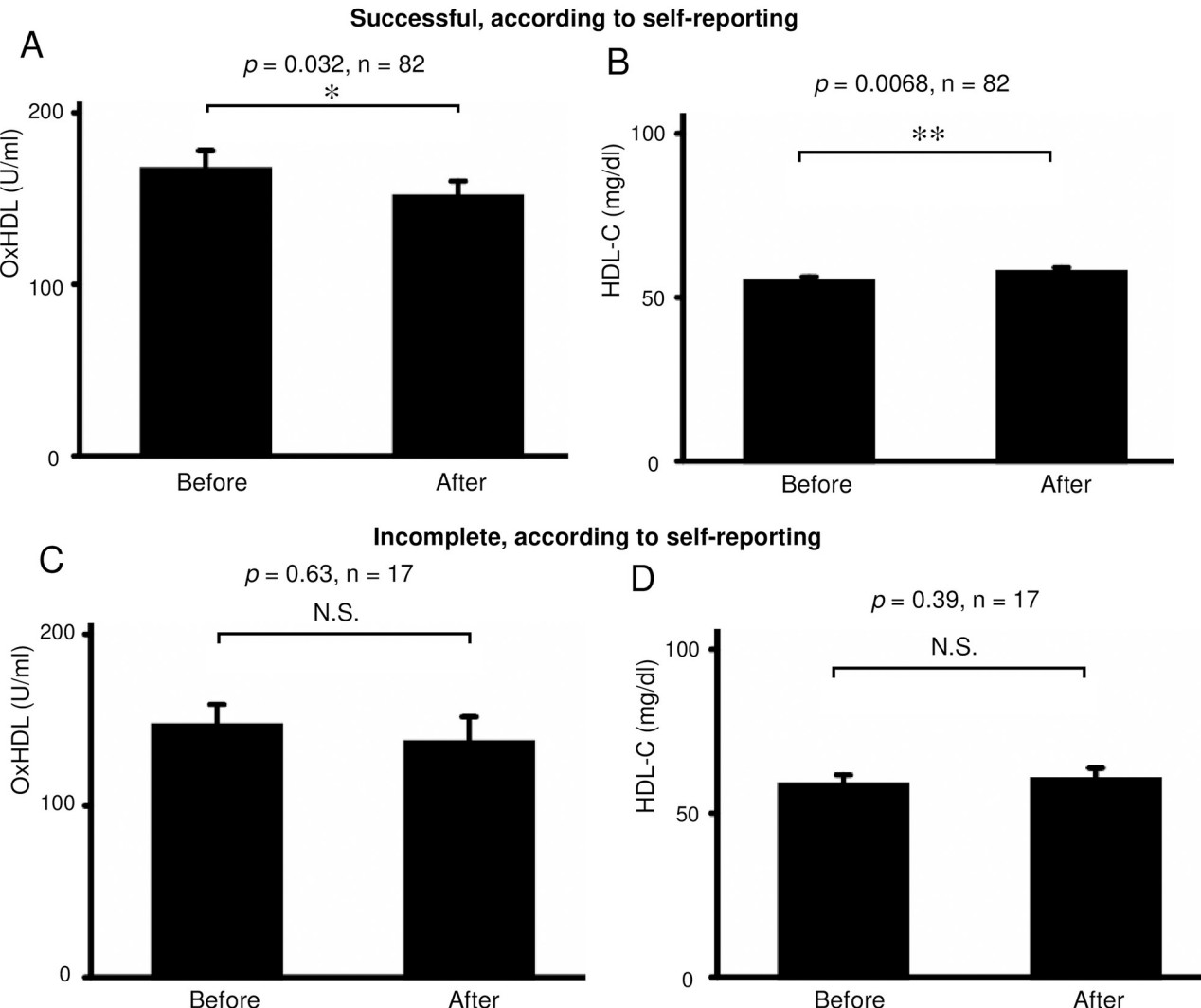

**Fig 3. Subgroup analysis of Fig 2.** Subjects were divided into two groups of successful and incomplete subjects based on the self-reports. In successful subjects, varenicline significantly decreased the serum concentration of oxHDL, but it significantly increased HDL-C (A, B). In incomplete subjects, these effects by varenicline were not statistically significant (C, D). * $p < 0.05$, ** $p < 0.01$. Bars: SEM. Abbreviations: HDL-C, high-density lipoprotein cholesterol; oxHDL, oxidized high-density lipoprotein.

subjects. The before-after change in oxHDL (ΔoxHDL) was compared with the before-after changes in various items; the comparison did not reveal clear associations. Nevertheless, ΔHDL-C had a positive association with change in total protein, cholinesterase, and red blood cell count. Abbreviations, quantity units, and normal ranges are listed in S1 Table.

## Relationship between before-after changes in oxHDL and exhaled CO

There was no significant association between ΔoxHDL and ΔCO in the total study population (n = 99) (Fig 7A). However, when subjects with a decrease in CO by ≥ 15 ppm after treatment with varenicline were selected, a close correlation between ΔoxHDL and ΔCO was noted (R = 0.42, $p = 0.0052$, n = 43) (Fig 7B). There was no significant association between ΔHDL-C and ΔCO (n = 99) (Fig 7C), even after the aforementioned selection (n = 43) (Fig 7D).

**Omission: exhaled CO at 3 months ≥ 10 ppm**

**Fig 4. After data omission from Fig 2.** Data of subjects in whom the treatment was objectively unsuccessful (exhaled CO at 3 months ≥ 10 ppm) were omitted (n = 93). (A) OxHDL was significantly decreased after smoking cessation with varenicline ($p = 0.0063$, n = 93). (B) HDL-C was significantly increased after smoking cessation with varenicline ($p = 0.011$, n = 93). * $p < 0.05$, ** $p < 0.01$. Bars: SEM. Abbreviations: CO, carbon monoxide; HDL-C, high-density lipoprotein cholesterol; oxHDL, oxidized high-density lipoprotein.

## Multiple regression analysis

In this multiple regression analysis using the data of subjects in Fig 7B or 7D (Table 2), the coefficient of determination ($R^2$) was 0.88. ΔoxHDL was significantly associated with ΔCO ($p = 0.022$, n = 43) (Table 2). ΔHDL-C was not significantly associated with ΔCO. Age was significantly associated with ΔCO independent of ΔoxHDL ($p < 0.0001$, n = 43). Sex did not demonstrate a significant association with ΔCO.

## Discussion

By measuring the serum concentration of oxHDL before and after smoking cessation with varenicline, we obtained the following findings. Firstly, although the baseline serum concentrations of oxHDL and HDL-C were closely correlated, they showed opposite changes (statistically significant decrease and increase, respectively) after smoking cessation. Secondly, although these statistical significances were also seen in the successful subgroup, they were not seen in the incomplete subgroup. Thirdly, routine clinical blood testing revealed a positive association between baseline oxHDL and baseline BUN, and a negative association between baseline oxHDL and baseline TG. Finally, although clear associations between ΔoxHDL and changes in parameters of typical clinical blood tests were not found, ΔoxHDL and ΔCO showed a positive association when we selected subjects in whom smoking cessation with varenicline was very effective (subjects with a decrease in CO by ≥ 15 ppm after treatment with varenicline).

Miki et al. recently reported that the decrease in oxHDL is associated with slowed progression of coronary artery calcification [14]. Here we found that oxHDL decreased after treatment with varenicline. Therefore, the present findings suggest that oxHDL is involved in the calcification of the coronary artery which is worsened by smoking. The definite explanations of the

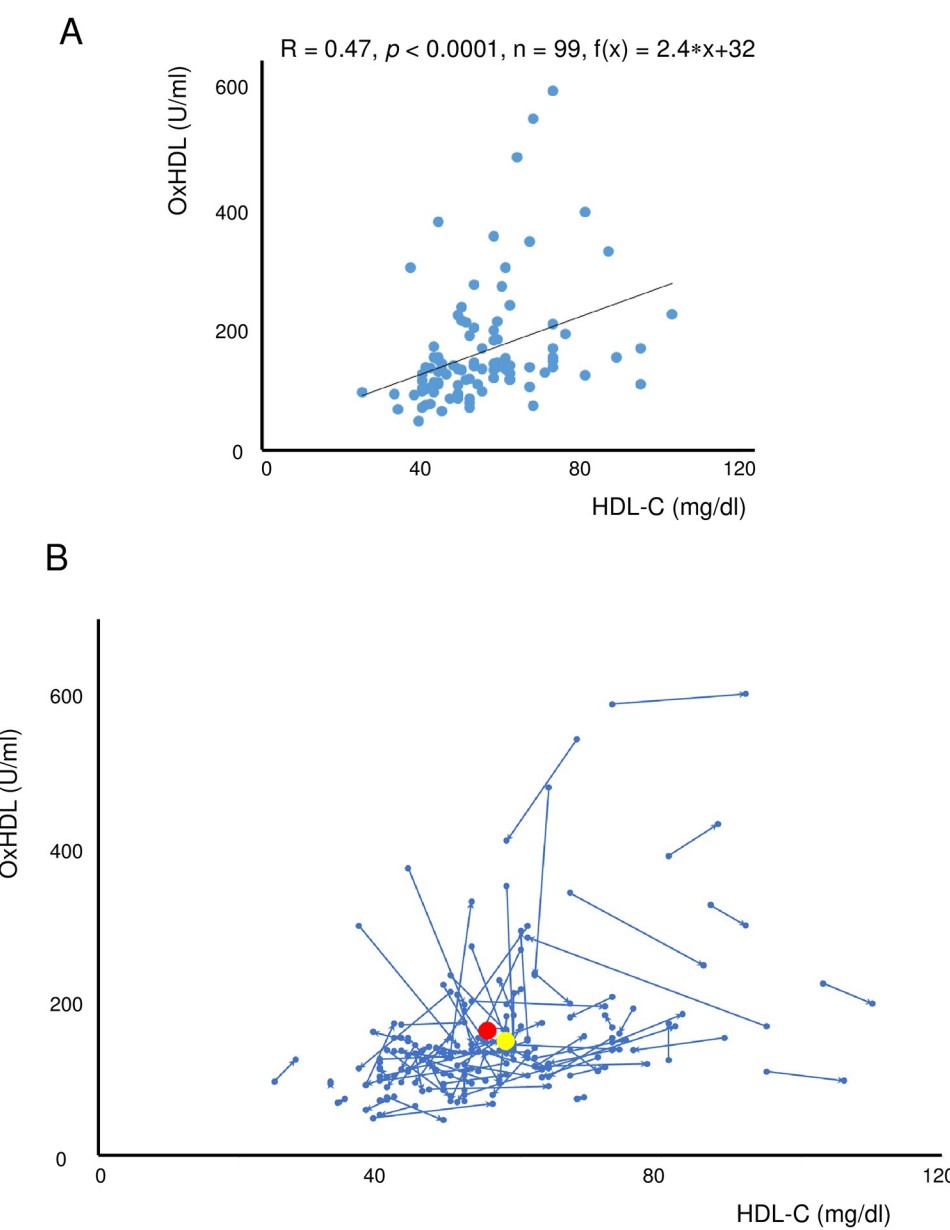

**Fig 5. Relationships between the serum levels of oxHDL and HDL-C in nicotine-dependent subjects.** (A) There was a significant correlation between the baseline serum levels of oxHDL and HDL-C (R = 0.47, $p < 0.0001$). (B) Arrow blots of serum oxHDL and HDL-C levels in nicotine-dependent subjects receiving varenicline for smoking cessation. The vector of average changes is shown from the red circle to the yellow circle. Abbreviations: HDL-C, high-density lipoprotein cholesterol; oxHDL, oxidized high-density lipoprotein.

cross-sectional positive correlation of HDL-C to oxHDL remain unclear, but for instance, in chronic exposure to smoking, a native HDL particle may be hypothesized to act as an acceptor to the smoking-induced oxidants as a previous study presumed a removal action of HDL particles to the smoking-induced oxidants [29]. In addition, it seems that high levels of HDL-C do not always mean protecting against coronary artery disease [30]. Furthermore, oxHDL has been used in animal experimental models of vascular injury [19]. Therefore, the decrease in oxHDL after smoking cessation may contribute to vascular health in abstainers.

| | Before (baseline) | After | | R between baseline data vs. OxHDL | vs. HDL | R between change data vs. ΔoxHDL | vs. ΔHDLC |
|---|---|---|---|---|---|---|---|
| BW | 65.0 ± 13.5 | 66.5 ± 14.2 | *** | -0.08 | -0.30 ** | -0.03 | -0.01 |
| BMI | 23.9 ± 4.2 | 24.4 ± 4.2 | *** | -0.08 | -0.29 ** | -0.04 | -0.01 |
| SBP | 127 ± 18 | 127 ± 17 | | 0.00 | -0.21 * | 0.01 | 0.05 |
| DBP | 77 ± 10 | 78 ± 11 | | 0.03 | -0.16 | -0.05 | 0.24 * |
| Pulse | 69.7 ± 12.9 | 70.7 ± 12.0 | | -0.07 | -0.14 | -0.01 | -0.09 |
| Tobaccos | 21.7 ± 7.9 | 1.3 ± 5.2 | *** | 0.07 | -0.13 | 0.10 | -0.02 |
| CO | 22.4 ± 12.8 | 4.9 ± 5.5 | *** | -0.02 | -0.10 | 0.09 | -0.18 |
| Pack-years | 41.7 ± 22.1 | N/A | | -0.03 | -0.18 | N/A | N/A |
| FMD | 5.01 ± 2.27 | 5.11 ± 2.15 | | -0.08 | -0.05 | 0.04 | -0.18 |
| **Routine blood tests** | | | | | | | |
| TP | 7.14 ± 0.47 | 7.19 ± 0.48 | | -0.02 | -0.13 | 0.16 | 0.40 ** |
| Alb | 4.29 ± 0.31 | 4.31 ± 0.30 | | 0.07 | -0.05 | 0.23 | 0.30 * |
| A/G | 1.53 ± 0.25 | 1.51 ± 0.23 | | 0.12 | 0.10 | 0.11 | -0.12 |
| GOT(AST) | 24.4 ± 9.2 | 25.4 ± 11.9 | | 0.11 | 0.18 | 0.15 | 0.20 |
| GPT(ALT) | 24.4 ± 13.6 | 27.0 ± 20.8 | | 0.08 | -0.01 | 0.19 | 0.14 |
| LDH | 189 ± 38 | 195 ± 41 | | 0.11 | 0.14 | 0.06 | 0.23 |
| γ-GTP | 50.2 ± 54.7 | 60.8 ± 76.0 | ** | 0.06 | -0.03 | 0.24 * | 0.13 |
| ChE | 333 ± 74 | 342 ± 69 | | -0.11 | -0.12 | 0.05 | 0.32 ** |
| TB | 0.54 ± 0.23 | 0.61 ± 0.20 | * | 0.03 | 0.04 | 0.10 | -0.01 |
| BUN | 14.0 ± 4.3 | 13.7 ± 3.6 | | 0.37 *** | 0.31 ** | -0.08 | 0.13 |
| CRTNN | 0.78 ± 0.15 | 0.79 ± 0.15 | | -0.11 | -0.23 * | -0.23 * | 0.00 |
| eGFR | 77.0 ± 14.8 | 75.9 ± 13.0 | | 0.08 | 0.14 | 0.25 * | 0.03 |
| Na | 141.3 ± 2.3 | 140.4 ± 2.0 | ** | -0.20 | -0.04 | -0.11 | -0.06 |
| K | 4.32 ± 0.38 | 4.29 ± 0.38 | | 0.24 * | 0.03 | 0.07 | 0.30 * |
| Cl | 105.4 ± 2.7 | 104.7 ± 2.6 | * | -0.05 | 0.10 | -0.15 | -0.14 |
| TC | 191.5 ± 35.1 | 199.0 ± 41.9 | | -0.21 | 0.08 | -0.06 | 0.13 |
| TG | 165.5 ± 107.1 | 171.8 ± 115.4 | | -0.26 ** | -0.45 *** | -0.19 | -0.25 * |
| LDLC | 113.4 ± 27.0 | 117.6 ± 27.6 | | -0.25 * | -0.13 | -0.14 | 0.09 |
| CRP | 0.146 ± 0.201 | 0.147 ± 0.252 | | -0.07 | -0.30 * | -0.05 | -0.19 |
| WBC | 7282 ± 2211 | 7171 ± 1736 | | -0.17 | -0.32 ** | 0.05 | 0.06 |
| Neu% | 55.6 ± 9.4 | 55.0 ± 7.2 | | -0.05 | -0.07 | -0.06 | 0.06 |
| Ly% | 32.5 ± 8.7 | 32.6 ± 6.9 | | 0.03 | 0.03 | 0.05 | 0.02 |
| Mono% | 8.0 ± 1.7 | 8.3 ± 2.1 | | 0.13 | 0.15 | -0.17 | -0.15 |
| Eos% | 2.7 ± 1.7 | 3.0 ± 1.8 | | -0.19 | -0.16 | -0.10 | -0.29 * |
| Baso% | 1.1 ± 0.4 | 1.1 ± 0.4 | | -0.05 | 0.07 | 0.11 | 0.05 |
| RBC | 457 ± 56 | 452 ± 50 | | -0.17 | -0.36 ** | 0.14 | 0.35 ** |
| Hb | 14.5 ± 1.4 | 14.3 ± 1.3 | | -0.05 | -0.29 * | 0.16 | 0.37 ** |
| Hct | 43.4 ± 4.4 | 42.6 ± 4.0 | | -0.13 | -0.32 ** | 0.15 | 0.35 ** |
| Plt | 23.2 ± 6.4 | 23.8 ± 6.5 | | -0.21 | -0.01 | 0.02 | -0.03 |
| HbA1c | 5.99 ± 0.82 | 6.15 ± 1.00 | *** | 0.17 | -0.02 | 0.08 | -0.01 |
| BS | 116.2 ± 39.3 | 112.3 ± 34.3 | | -0.03 | -0.08 | 0.01 | 0.11 |
| | | | | * $p < 0.05$, ** $p < 0.01$, *** $p < 0.001$ | | | |

**Fig 6. Changes in various parameters and association with oxHDL or HDL-C.** Abbreviations, quantity units, and normal ranges are listed in S1 Table.

The $p$-value on the changes in the serum concentrations of oxHDL after the smoking cessation was still large (0.034) with n = 99. It was almost in a marginal range although it meant significant. Therefore, we evaluated successful subgroups. By omitting the data of objectively unsuccessful subjects with exhaled CO at 3 months ≥ 10 ppm, this $p$-value decreased to 0.0063 (n = 93). Thus, this effect was thought to be associated with the effectiveness of smoking cessation.

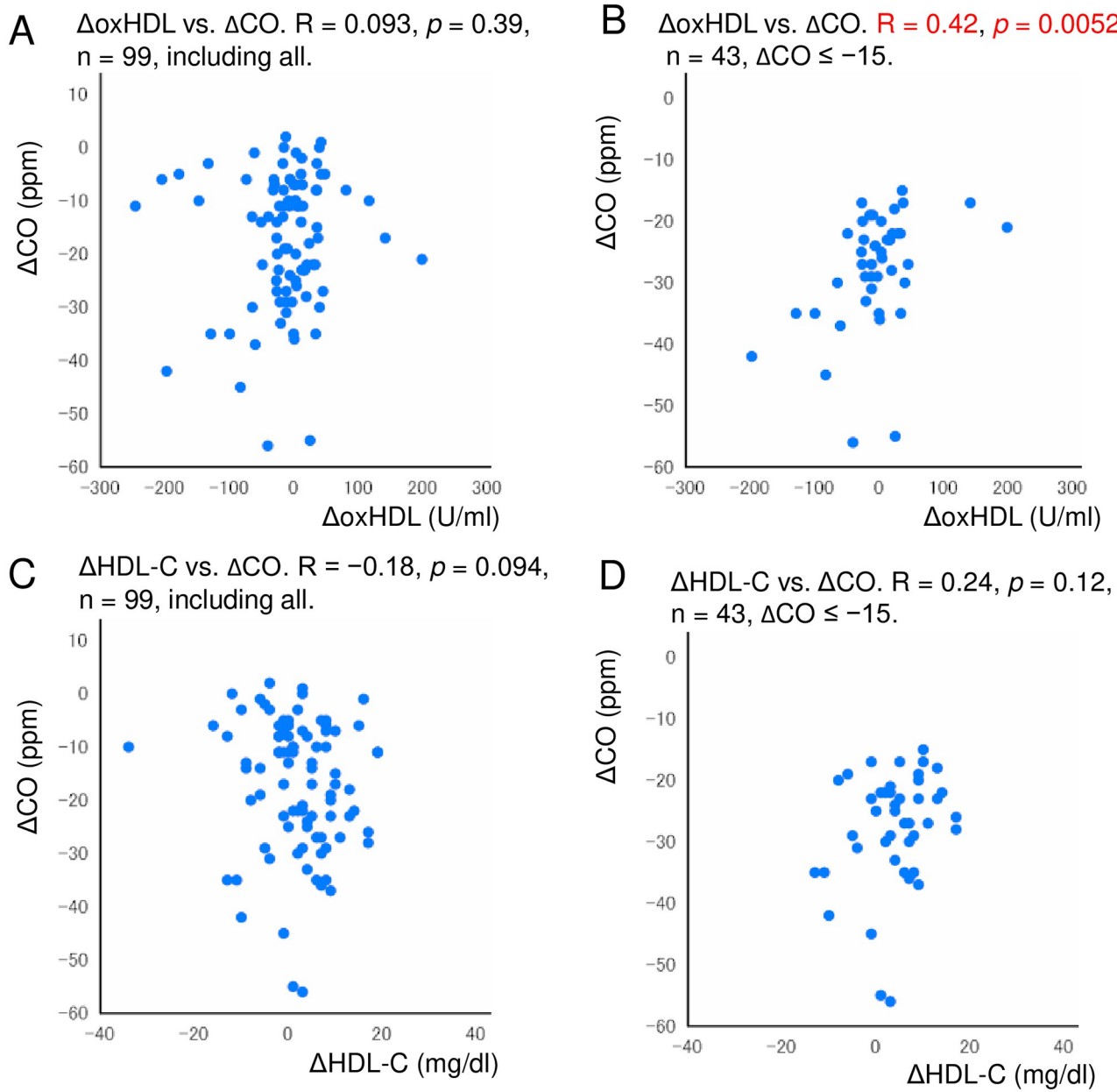

**Fig 7. Relationships between changes in exhaled CO and changes in oxHDL or HDL-C levels in subjects receiving varenicline for smoking cessation.** (A) There was no significant association observed between ΔoxHDL and ΔCO in total (n = 99). (B) When subjects with a decrease in CO by ≥ 15 ppm after treatment with varenicline were selected, a close correlation between ΔoxHDL and ΔCO was observed (R = 0.42, $p < 0.01$, n = 43). (C) There was no significant association noted between ΔHDL-C and ΔCO (n = 99). (D) There was no significant association recorded between ΔHDL-C and ΔCO even after the aforementioned selection (n = 43). Abbreviations: CO, carbon monoxide; HDL-C, high-density lipoprotein cholesterol; oxHDL, oxidized high-density lipoprotein.

The pathophysiological significance of oxHDL remains unclear. Therefore, we evaluated the relationships between oxHDL and various parameters used in typical clinical practice. The baseline oxHDL showed a positive association with BUN, and a negative association with TG. HDL-C has been associated with renal dysfunction in patients with heart failure [31].

**Table 2. Multiple regression analysis by forced entry of four explanatory variables (ΔoxHDL, ΔHDL-C, age, and sex) for ΔCO in subjects in whom smoking cessation with varenicline was very effective (n = 43, ΔCO ≤ −15 ppm).**

| Explanatory variable | R-value in univariate analysis | p-value in univariate analysis | Standardized regression coefficient | p-value in multiple regression analysis |
|---|---|---|---|---|
| ΔoxHDL | 0.42 | 0.0052** | 0.14 | 0.022* |
| ΔHDL-C | 0.24 | 0.12 | −0.01 | 0.89 |
| Age | 0.19 | 0.22 | −0.81 | <0.0001*** |
| Sex | 0.022 | 0.89 | −0.11 | 0.38 |

* $p < 0.05$,

** $p < 0.01$,

*** $p < 0.001$

Abbreviations: CO, carbon monoxide; HDL-C, high-density lipoprotein cholesterol; oxHDL, oxidized high-density lipoprotein

Moreover, HDL-C is negatively associated with TG in smokers [32]. Similar associations may also be observed for oxHDL. Further investigation is warranted to confirm these findings.

Similarly, we evaluated the relationships between ΔoxHDL and changes in various parameters. After omitting data of subjects with a smaller decrease in exhaled CO (< 15 ppm), we found that ΔoxHDL and Δ(exhaled CO) exhibited a close positive association. Through this omission, we could select subjects in whom smoking cessation with varenicline was very effective. Subjects in whom smoking cessation was unsuccessful or incompletely successful may show a slight decrease in exhaled CO. Also, relatively light smokers may also demonstrate a limited reduction in exhaled CO.

The multiple regression analysis identified ΔoxHDL as a significant explanatory variable for ΔCO in subjects in whom smoking cessation was very effective. Although age did not have a significant association with ΔCO in the univariate analysis, it was a significant explanatory variable for ΔCO in the multiple regression analysis. The years of smoking were supposed to reflect the absolute value of ΔCO after eliminating the factor of ΔoxHDL. Univariate and multiple regression analyses demonstrated that ΔHDL-C was not significantly associated with ΔCO. Therefore, oxHDL and HDL-C were independent of the effectiveness of smoking cessation with varenicline. The problem of multicollinear variables between ΔoxHDL and ΔHDL-C is negligible because the R-value was < 0.8 [33]. Although we did not take into account changes in lifestyle (exercise, diet) after smoking quit in this multiple regression analysis, consideration of this lifestyle change factor may be important for future studies.

In an ultrasound examination, flow-mediated vasodilation (FMD) is a marker of vascular endothelial function [26]. We reported that this index was increased after successful smoking cessation with varenicline, and oxidative stress appeared to be involved in this process [27]. There was no significant association between baseline oxHDL and baseline FMD, as well as between ΔoxHDL and ΔFMD. Although both oxHDL and FMD have been reported to associate with atherosclerosis, these two factors might not be linked. The oxidized HDL has been reported to appear in pathophysiologic conditions with dysfunction of HDL, assuming that oxHDL can be a measure of dysfunctional HDL [11, 17, 34, 35]. Thus, oxHDL can be a risk marker for atherosclerosis and cardiovascular disease.

Previously, we evaluated serum derivatives of reactive oxygen metabolites as markers of oxidative stress in smoking cessation subjects. We reported that they significantly decreased after smoking cessation with varenicline [27]. Other markers of oxidative stress associated with smoking include urinary 8-hydroxy-2'-deoxyguanosine and 8-iso-prostaglandin F2α [36–39]. It has been reported that these markers are decreased after smoking cessation [36, 37]. Systematic

evaluation and meta-analysis of these markers of oxidative stress are warranted. Understanding of these markers and the exhaled CO reported in this study may be a method of evaluation.

Experimental studies report that varenicline itself affected oxidative stress in rats [40, 41]. According to their "chronic" experiments, varenicline significantly decreased antioxidants of superoxide dismutase, catalase, glutathione, and glutathione peroxidase levels, and increased oxidant levels comprising malondialdehyde and myeloperoxidase. Oguz et al. reported that the "acute" usage of varenicline (mimicking the human use for smoking cessation) significantly decreased rats' superoxide dismutase in kidney and testis tissues [40]. From our study, the reduction of oxidation to HDL via smoking cessation is wholly thought to be superior to the increased tendency to oxidation by varenicline itself. Smoking induces oxidation of proteins and lipids [42, 43], and oxHDL stems from oxidative modification of HDL (containing apoA-I). Thus, the current study suggests that the peroxidation content of HDL is also reduced following smoking cessation with the use of varenicline. Together, smoking cessation without the help of varenicline might be better for vascular health.

## Conclusions

In conclusion, although there was a close relationship between the baseline serum concentrations of oxHDL and HDL-C, smoking cessation with varenicline decreased oxHDL and increased HDL-C. This effect on oxHDL was thought to be associated with the effectiveness of smoking cessation.

## Supporting information

**S1 Table. More information of Fig 6.** Abbreviations, quantity units, and normal ranges are listed.
(DOCX)

**S1 File.**
(XLSX)

## Acknowledgments

We thank all the participants of this study.

## Author Contributions

**Conceptualization:** Akira Umeda, Toru Kato, Kazuhiko Kotani.

**Data curation:** Akira Umeda, Kazuhiko Kotani.

**Formal analysis:** Akira Umeda, Kazuhiko Kotani.

**Funding acquisition:** Kazuhiko Kotani.

**Investigation:** Akira Umeda, Kazuya Miyagawa, Atsumi Mochida, Hiroshi Takeda, Yoshiyuki Ohira, Toru Kato, Yasumasa Okada, Kazuhiko Kotani.

**Methodology:** Akira Umeda, Toru Kato, Kazuhiko Kotani.

**Project administration:** Kazuya Miyagawa, Hiroshi Takeda, Kazuhiko Kotani.

**Supervision:** Hiroshi Takeda.

**Validation:** Yasumasa Okada.

**Writing – original draft:** Akira Umeda.

**Writing – review & editing:** Kazuya Miyagawa, Atsumi Mochida, Hiroshi Takeda, Yoshiyuki Ohira, Toru Kato, Yasumasa Okada, Kazuhiko Kotani.

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
