## [Decision Letter · Decision Letter 0]

9 Aug 2022

PONE-D-22-07258Effects of smoking cessation using varenicline on the serum concentrations of oxidized high-density lipoprotein: Comparison with high-density lipoprotein cholesterolPLOS ONE

Dear Dr. Umeda,

Thank you for submitting your manuscript to PLOS ONE. After careful consideration, we feel that it has merit but does not fully meet PLOS ONE’s publication criteria as it currently stands. Therefore, we invite you to submit a revised version of the manuscript that addresses the points raised during the review process.

We look forward to receiving your revised manuscript.

Kind regards,

Laura Calabresi

Academic Editor

PLOS ONE

Journal Requirements:

Reviewers' comments:

Reviewer's Responses to Questions

**Comments to the Author**

1. Is the manuscript technically sound, and do the data support the conclusions?

Reviewer #1: Partly

Reviewer #2: Yes

2. Has the statistical analysis been performed appropriately and rigorously? 

Reviewer #1: N/A

Reviewer #2: Yes

3. Have the authors made all data underlying the findings in their manuscript fully available?

Reviewer #1: Yes

Reviewer #2: Yes

4. Is the manuscript presented in an intelligible fashion and written in standard English?

Reviewer #1: Yes

Reviewer #2: Yes

5. Review Comments to the Author

Reviewer #1: In this work the authors demonstrated that smoking cessation with varenicline decreased oxHDL and increased HDL-C and the effect on oxHDL was thought to be associated with the effectiveness of smoking cessation. Even the work is well conducted, it has several limitations that have to be addressed:

1. It seems that oxHDL are positively correlated with serum levels of HDL-C. How the authors explain this correlation? It seems that subject with high level of HDL-C, potentially more protected, have also high level of dysfunctional oxHDL. Authors should discuss this point.

2. Figure 5A. Add the linear correlation in the graph will clarify the message

3. Do the authors take into account change in life-style (exercise, diet) after smoking quite in multiple regression analysis? This is an important point that can influence HDL-C concentration.

4. The study completely lack of functional assays to determine HDL functionality which are essential to support conclusions.

5. It is unclear if the correlation with oxHDL and FMD was related to a previous study, it is important add FMD evaluation also for this study.

Reviewer #2: This is an interesting and well-presented article on the effects of smoking cessation with varenicline on the levels of oxHDL.

I have few comments that could improve the manuscript:

1)The determination of oxHDL is performed using a monoclonal antibody against oxidized apoA-I. Is it known what type of oxidation in apoA-I is determined?

2)The role of oxidized apoA-I in atherosclerosis and cardiovascular disease should be discussed.

3)How do the authors speculate that varenicline could affect the levels of oxHDL? Is there an effect of varenicline in myeloperoxidase (MPO) mass or activity in subjects?

4)Is lipid peroxidation content of HDL also reduced following smoking sensation?

5)Introduction: “oxHDL has been used in animal models of vascular injury”. What does this mean? What was the use of ox-HDL in the particular animal model?

6. PLOS authors have the option to publish the peer review history of their article (what does this mean?). If published, this will include your full peer review and any attached files.

Reviewer #1: No

Reviewer #2: No

---

## [Author Response · Author response to Decision Letter 0]

21 Sep 2022

Dear Reviewers of PLOS ONE,

Hello. Thank you very much for reviewing our paper entitled, “Effects of smoking cessation using varenicline on the serum concentrations of oxidized high-density lipoprotein: Comparison with high-density lipoprotein cholesterol”.

We would like to respond to your comments as follows.

Reviewer #1: 

(Comment)

In this work the authors demonstrated that smoking cessation with varenicline decreased oxHDL and increased HDL-C and the effect on oxHDL was thought to be associated with the effectiveness of smoking cessation. Even the work is well conducted, it has several limitations that have to be addressed:

1. It seems that oxHDL are positively correlated with serum levels of HDL-C. How the authors explain this correlation? It seems that subject with high level of HDL-C, potentially more protected, have also high level of dysfunctional oxHDL. Authors should discuss this point.

(Response)

The definite explanations of the cross-sectional positive correlation of HDL-C to oxHDL remain unclear, but for instance, in chronic exposure to smoking, a native HDL particle may be hypothesized to act as an acceptor to the smoking-induced oxidants as a previous study presumed a removal action of HDL particles to the smoking-induced oxidants [29]. In addition, it seems that high levels of HDL-C do not always mean protecting against coronary artery disease [30].

We wrote these sentences in the middle of the 2nd paragraph of the Discussion (Line 255-61).

(Comment)

2. Figure 5A. Add the linear correlation in the graph will clarify the Message

(Response)

We added the linear correlation in the graph in Figure 5A.

(Comment)

3. Do the authors take into account change in life-style (exercise, diet) after smoking quite in multiple regression analysis? This is an important point that can influence HDL-C concentration.

(Response)

Although we did not take into account changes in lifestyle (exercise, diet) after smoking quit in this multiple regression analysis, consideration of this lifestyle change factor may be important for future studies. (We wrote this in the last of the 6th paragraph in the Discussion. Line 297-9)

(Comment)

4. The study completely lack of functional assays to determine HDL functionality which are essential to support conclusions.

(Response)

The oxidized HDL has been reported to appear in pathophysiologic conditions with dysfunction of HDL, assuming that oxHDL can be a measure of dysfunctional HDL [11, 17, 34, 35]. Thus, oxHDL can be a risk marker for atherosclerosis and cardiovascular disease. We wrote these sentences in the 7th paragraph of the Discussion (Line 307-10).

(Comment)

5. It is unclear if the correlation with oxHDL and FMD was related to a previous study, it is important add FMD evaluation also for this study.

(Response)

There was no significant association between baseline oxHDL and baseline FMD, as well as between ΔoxHDL and ΔFMD. Although both oxHDL and FMD have been reported to associate with atherosclerosis, these two factors might not be linked. (We wrote this in the 7th paragraph of the Discussion. Line 305-7) 

Reviewer #2: 

(Comment)

This is an interesting and well-presented article on the effects of smoking cessation with varenicline on the levels of oxHDL. I have few comments that could improve the manuscript:

1)The determination of oxHDL is performed using a monoclonal antibody against oxidized apoA-I. Is it known what type of oxidation in apoA-I is determined?

(Response)

Originally, the assay of oxHDL used in the study was based on the H2O2-induced oxidative modification of apoA-I [11]. We wrote this in the “Measurement of oxHDL and HDL-C” paragraph of Methods (Line 134-5). 

(Comment)

2)The role of oxidized apoA-I in atherosclerosis and cardiovascular disease should be discussed.

(Response)

The oxidized HDL has been reported to appear in pathophysiologic conditions with dysfunction of HDL, assuming that oxHDL can be a measure of dysfunctional HDL [11, 17, 34, 35]. Thus, oxHDL can be a risk marker for atherosclerosis and cardiovascular disease. We wrote these sentences in the last of the 7th paragraph of the Discussion (Line 307-10).

(Comment)

3)How do the authors speculate that varenicline could affect the levels of oxHDL? Is there an effect of varenicline in myeloperoxidase (MPO) mass or activity in subjects?

(Response)

Experimental studies report that varenicline itself affected oxidative stress in rats [40, 41]. According to their “chronic” experiments, varenicline significantly decreased antioxidants of superoxide dismutase, catalase, glutathione, and glutathione peroxidase levels, and increased oxidant levels comprising malondialdehyde and myeloperoxidase. Oguz et al. reported that the “acute” usage of varenicline (mimicking the human use for smoking cessation) significantly decreased rats’ superoxide dismutase in kidney and testis tissues [40]. From our study, the reduction of oxidation to HDL via smoking cessation is wholly thought to be superior to the increased tendency to oxidation by varenicline itself. We wrote these sentences in the last paragraph of the Discussion (Line 321-9). 

(Comment)

4)Is lipid peroxidation content of HDL also reduced following smoking sensation?

(Response)

Smoking induces oxidation of proteins and lipids [42, 43], and oxHDL stems from oxidative modification of HDL (containing apoA-I). Thus, the current study suggests that the peroxidation content of HDL is also reduced following smoking cessation with the use of varenicline. Together, smoking cessation without the help of varenicline might be better for vascular health. We wrote these sentences in the last paragraph of the Discussion (Line 329-33).

(Comment)

5)Introduction: “oxHDL has been used in animal models of vascular injury”. What does this mean? What was the use of ox-HDL in the particular animal model?

(Response)

He et al. demonstrated that apoA-I mimetic peptide 4F inhibited oxHDL-induced dysfunction of endothelial repairing in a mice model of vascular electric injury [19]. We wrote this in the last of the second paragraph in the Introduction (Line 79-81). 

Changed portions are indicated in red. 

We made some tiny corrections using the software “Grammarly”. 

We consistently used the abbreviation of apolipoprotein A-1 (apoA-I). (Line 71 and throughout the paper)

Fig. 6. Row 1 column 5. “R between before data” was changed to “R between baseline data”. 

We hope you will feel our paper interesting and accept it for publication. 

Sincerely,

21 Sep 2022

Akira Umeda, M.D., Ph.D.

---

## [Decision Letter · Decision Letter 1]

3 Nov 2022

Effects of smoking cessation using varenicline on the serum concentrations of oxidized high-density lipoprotein: Comparison with high-density lipoprotein cholesterol

PONE-D-22-07258R1

Dear Dr. Umeda,

We’re pleased to inform you that your manuscript has been judged scientifically suitable for publication and will be formally accepted for publication once it meets all outstanding technical requirements.

Kind regards,

Donovan Anthony McGrowder, PhD., MA., MSc

Academic Editor

PLOS ONE

Additional Editor Comments:

Dear Dr. Umeda,

The manuscript entitled “Effects of smoking cessation using varenicline on the serum concentrations of oxidized high-density lipoprotein: Comparison with high-density lipoprotein cholesterol” was revised in accordance with the reviewers’ comments and is provisionally accepted pending final checks for formatting and technical requirements.

Regards,

Dr. Donovan McGrowder (Academic Editor)

---

## [Editor Report · Acceptance letter]

18 Nov 2022

PONE-D-22-07258R1 

Effects of smoking cessation using varenicline on the serum concentrations of oxidized high-density lipoprotein: Comparison with high-density lipoprotein cholesterol 

Dear Dr. Umeda:

I'm pleased to inform you that your manuscript has been deemed suitable for publication in PLOS ONE. Congratulations! Your manuscript is now with our production department. 

Kind regards, 

on behalf of

Dr. Donovan Anthony McGrowder 

Academic Editor

PLOS ONE